# Agarwood in the Modern Era: Integrating Biotechnology and Pharmacology for Sustainable Use

**DOI:** 10.3390/ijms26178468

**Published:** 2025-08-30

**Authors:** Aqsa Baig, Adeel Akram, Ming-Kuem Lin

**Affiliations:** 1Graduate Institute of Biological Science and Technology, College of Life Science, China Medical University, Taichung 40402, Taiwan; u112301141@cmu.edu.tw; 2Department of Chinese Pharmaceutical Sciences and Chinese Medicine Resources, College of Chinese Medicine, China Medical University, Taichung 40402, Taiwan; u113047141@cmu.edu.tw

**Keywords:** agarwood, *Aquilaria*, authentication, medicinal plant, DNA barcoding, sesquiterpene, chromone

## Abstract

Agarwood, valued for its resin, has long been used in perfumery, incense, and traditional medicine. Its resin is primarily derived from species of *Aquilaria* and is produced through a still-unknown process in response to biotic or abiotic stress. Concerns regarding agarwood’s sustainability and conservation have emerged because of the substantial loss of natural resources due to overharvesting and illegal trade. To address these concerns, artificial techniques are being used to produce agarwood. The mechanism underlying agarwood production must be elucidated to enhance yield. The authentication of agarwood species is challenging because of morphological similarities between pure and hybrid *Aquilaria* species. Techniques such as DNA barcoding, molecular marker assessment, and metabolomics can ensure accurate identification, facilitating conservation. Artificial intelligence and machine learning can support this process by enabling rapid, automated identification on the basis of genetic and phytochemical data. Advances in resin induction methods (e.g., fungal inoculation) and chemical induction treatments are improving yield and quality. Endophytic fungi and bacteria promote resin production at minimal harm to the tree. Agarwood’s pharmacological potential—antimicrobial, anti-inflammatory, and anticancer effects—has driven research into bioactive compounds such as sesquiterpenes and flavonoids for the development of novel drugs. This systematic review synthesized current evidence on species authentication, induction techniques, and pharmacological properties. The findings may guide future research aimed at ensuring sustainable use and enhancing the medicinal value of agarwood.

## 1. Introduction

### 1.1. Importance of Agarwood

Agarwood, also known as oud, aloeswood, or gharuwood, is a fragrant dark resinous wood formed in the heartwood of *Aquilaria* trees, particularly when they are wounded or infected by a fungus. Agarwood, formed due to microbial infection, secretes a dark, aromatic resin, which is produced through secondary metabolism by the *Aquilaria* tree. Agarwood is used in incense, perfumes, cosmetics, and medicine and is known by various names such as “agar” (in various South Asian languages), “Oudh” (in Arabic), and “jinkoh” (in Japanese).

Agarwood is among the most expensive natural products, with a kilogram of pure agarwood costing as much as US$30,000 [1]. The cost of agarwood varies globally; the price per kilogram of wood chips ranges from US$20 to US$6000 depending on quality, whereas the cost of the wood itself is US$10,000 per kilogram. An estimated US$6–US$8 billion is traded annually on the global agarwood market, yet many transactions are unrecorded [2].

In Asia, agarwood has long been used in religious rituals and Traditional Chinese Medicine. Agarwood leaves are used in religious ceremonies, perfumery, and aromatherapy. Regarding medicinal value, agarwood contains various compounds with neuroprotective, sedative, antibacterial, anti-inflammatory, and antioxidative properties [3,4].

Healthy *Aquilaria* trees can only generate a limited quantity of agarwood before sustaining severe damage or becoming overrun by microbes. Wild agarwood has become scarce because of overharvesting [5]. In 2004, a list of all *Aquilaria* species was included in Appendix A of the CITES (Convention on International Trade in Endangered Species of Wild Fauna and Flora) treaty [6]. In response to external stress, healthy *Aquilaria* produces sesquiterpenes (C_15_) and 2-(2-phenylethyl) chromones, which accumulate over time to form high-quality resin. This slow process makes natural agarwood increasingly rare [7]. Thus, research efforts have been directed toward the large-scale production of agarwood through artificial means.

### 1.2. Challenges in Authenticating and Manufacturing Agarwood

Resin quality is key to agarwood authenticity. However, to the best of our knowledge, no standardized system is available for grading resin quality. DNA barcoding and related molecular tools are essential for accurately differentiating species with overlapping morphology, ensuring product authenticity, and reducing adulteration. Standardization is difficult to achieve because quality depends on fragrance, wood density, and resin content, which vary across batches. They provide a scientific basis for quality standardization, support conservation and regulatory efforts, and enable early species verification to prevent financial loss from cultivating non-resin-producing plants. Natural agarwood can be accurately differentiated from synthetic agarwood only with the help of sophisticated and inaccessible procedures such as spectroscopy or DNA analysis [8]. Resin production through artificial stimulation of cultivated trees remains challenging. Stressors such as physical injury or fungal inoculation do not guarantee the production of high-quality resin [9]. Even under stress, not all trees produce agarwood, making large-scale production challenging. Resin yield and quality vary depending on climate, tree genetics, and induction technique [10].

The market for agarwood and its derivatives, particularly agarwood oil, is substantial, with top-grade products commanding prices higher than that of gold [11]. Authenticity certification not only preserves market integrity but also ensures fair-trade and consumer protection [12].

The trade of agarwood is governed by various legal frameworks aimed at protecting consumers and ensuring ethical sourcing [13]. Ethical sourcing of agarwood not only supports local communities involved in its production but also adheres to international conservation efforts [14]. Advances in chemical analysis and DNA profiling techniques are necessary for accurate authentication of the origin, species, and quality of agarwood [15]. Furthermore, improvements in inoculation strategies and other resin induction methods are essential for ensuring reliable and sustainable agarwood production. Overcoming these obstacles in agarwood production requires balancing technology, ecofriendly cultivation, and quality control [5].

## 2. Species Authentication and DNA Barcoding Technology

The identification of *Aquilaria* spp. is a complex and laborious process. First, samples from different places are gathered to determine the primary morphological traits of a given species, especially species common to China, Malaysia, and Southeast Asia. Morphological, anatomical, and genetic methods are used to identify *Aquilaria* sp., from which agarwood is derived [16]. The morphological identification of a plant species relies on the characteristics of its fruit as well as its size and leaves. An anatomical assessment of the plant’s cellular organization and resin ducts is also required. Molecular approaches such as DNA barcoding on the basis of specific gene sections—for example, *maturase K* (*matK*), ribulose−1, 5-bisphosphate carboxylase/oxygenase large subunit (*rbcL*), or internal transcribed spacer—have been used to accurately differentiate between species [17]. With the current development of molecular and bioinformatics tools, DNA barcoding of agarwood (*Aquilaria* and *Gyrinops* spp.) can reliably provide information on species identification and authentication. The most commonly used loci for agarwood authentication are *matK*, *rbcL*, ITS, and *trnH–psbA*, which have proven discriminatory power in differentiating closely related *Aquilaria* species (e.g., *A. malaccensis*, *A. crassna*, *A. sinensis*) that are often morphologically similar. DNA barcoding is commonly used across various taxonomic levels (Figure 1) [18]. A study focusing on IUCN Red-listed *Aquilaria* species evaluated five DNA barcode loci—ITS, *matK*, *psbA-trnH*, *rbcL*, and *trnL-trnF*—both individually and in various combinations. The findings demonstrated that the combination of ITS2, *psbA-trnH*, and *matK* provided 100% species-level discrimination, whereas individual loci showed lower accuracy, ranging from 60% to 85% [19]. DNA identification of agarwood species is crucial for accurately and rapidly determining the correct variety. First, only genuine agarwood trees can produce resin, making it essential to confirm whether the tree belongs to the agarwood species. Second, identifying the exact variety is also important. Although there is a certain correlation between the species of agarwood and the quality of its aroma, the relationship is not absolute, as aroma quality can be influenced by external environmental factors and is also subject to individual perception. Nevertheless, it is undeniable that non-agarwood trees will not produce agarwood resin. Advanced techniques such as super-barcoding and meta-barcoding offer improved accuracy in identifying species at lower taxonomic levels but are mainly used to confirm results from standard single- or multilocus barcoding. Mini-barcoding is increasingly preferred for analyzing degraded DNA samples, allowing broader application in plant identification. Chemical profiling of sesquiterpenes is another technique occasionally used to differentiate between species with similar exterior traits [20].

## 3. Identification of *Aquilaria* sp.

*Aquilaria* sp. are identified through sequence analysis, genetic distance calculation, and phylogenetic tree construction. BLAST (version 2.17.0) is commonly used to align sequences and validate matches against the GenBank database. The TaxonDNA software (v.1.10) is used to calculate the rate of species identification, whereas the Figtree (version 1.4.3) and R (version 4.0.0) software are used to visualize and refine the construction of phylogenetic trees. DNA barcoding is a vital tool for species identification, conservation, quality control, research, and legal compliance [21,22].

In DNA barcoding, a short, species-specific genetic sequence is used to differentiate between plant species. This technique is widely used for taxonomic and forensic analyses, medicinal plant verification, and endangered species authentication [23]. It is extensively used for identifying *Aquilaria*. Standardized DNA segments are used in DNA barcoding for rapid and accurate species identification [24]. Although DNA barcoding is essential for distinguishing one *Aquilaria* sp. from another, different barcode regions may be required for different materials. DNA barcoding enhances identification accuracy when combined with traditional methods and used on appropriately collected samples. Standard methods cannot be used to trace the species origin of resinous products, but barcode sequences can help identify the source species of commercially sold agarwood [25]. Reported that *Aquilaria sinensis* was closely related to *Aquilaria yunnanensis* when chloroplast DNA barcodes were used and to *Aquilaria crassna* when both nuclear and chloroplast barcodes were used. In Southeast Asia, *Aquilaria malaccensis* is the primary source of premium agarwood. Thus, reliable tracking and authentication are required for this species [26]. DNA barcoding, based on DNA regions such as matK and rbcL, in chloroplast DNA can be used to effectively identify agarwood species.

Molecular markers, such as single-nucleotide polymorphisms and simple sequence repeats, can aid in agarwood species identification [27]. Generated DNA sequences are compared with reference databases or phylogenetic trees to accurately determine the species or variety [28]. DNA barcoding enables the determination of taxonomic position, supports efforts toward species conservation, and facilitates the authentication of agarwood in the market. Furthermore, it ensures sustainable use and protection of agarwood resources while preventing the exploitation of endangered species.

## 4. Assessment of Agarwood Authenticity

Agarwood’s distinct scent and traits are attributable to resinous chemicals, particularly sesquiterpenes and chromones, whose quantity and composition determine the quality of the wood [29]. Agarwood from different *Aquilaria* species and different geographic origins can vary significantly in value, uses, and market demand. These differences are mainly due to variations in resin composition, aroma profile, and cultural preferences, which directly affect grading and pricing. Resin accumulates as the tree responds to infection, with higher-quality agarwood containing more resin. Factors such as location, fungal species, and soil strongly influence resin production, affecting the quality, scent, and market value of the agarwood [30]. Accurate authentication of agarwood resin now combines traditional analyses with modern technologies. Recent advances in chemical profiling, AI, and machine vision have enhanced precision and objectivity. Table 1 summarizes these methods, key principles, findings, and applications. Characteristic chemicals in agarwood, identified through liquid extraction surface analysis (LESA)–mass spectrometry (MS), include 2-(2-phenylethyl) chromones (*m*/*z* 319.1), which can facilitate the rapid identification of agarwood [31]. LESA–MS can effectively analyze agarwood samples from various sources. The distinct woody scent of agarwood results from sesquiterpenes, such as eudesmane and agarofuran. Chromones, particularly 2-(2-phenylethyl) chromones, imbue agarwood with fruity, sweet, and balsamic notes. Phenolic compounds, including flavonoids, enhance the stability and durability of resin by affecting its oxidative characteristics [32,33]. Higher-quality agarwood has a more complex and potent chemical composition, which affects its grade and market value [34].

Essential oil fragrances are identified using electronic sensors or by a human evaluator through sensory evaluations [35]. Artificial intelligence (AI) and machine learning (ML) models are emerging tools in agarwood research for species identification, resin detection, and quality assessment. Deep learning segmentation has been used to detect resin zones in cross-sectional images and guide computer numerical control (CNC) machines in automatically removing resin [36]. ML-based machine vision can help identify resinous parts of an agarwood tree. Additionally, ML techniques can be applied to data obtained through Fourier transform infrared spectroscopy and gas chromatography (GC)–MS to classify agarwood species and predict resin quality; this approach facilitates a rapid and objective phytochemical analysis. Electronic sensors, such as electronic tongues or noses, enable objective, reproducible measurements of fragrance (Table 1). Both sensor-based and ML-based methods are crucial for determining the quality of essential oils [37].

**Table 1 ijms-26-08468-t001:** Overview of scientific and AI-based techniques used in the authentication and quality assessment of agarwood resin.

Method/Technique	Key Factor	Findings	Implications for Authentication
Anatomical analysis	Phloem	Presence of resin in phloem [34]	Non-chemical, microscopic test for identifying fake or resin-coated wood
Deep learning segmentation method	Segmentation of cross-sectional images; control of a CNC machine with G-code script	Automated CNC-guided resin removal through accurate resin boundary detection [37]	Machine vision method for identifying genuine resinous parts
Metabolite profiling through GC–MS and LC–MS	Discrimination between wild agarwood and cultivated agarwood	Difference in biochemical signature between wild agarwood and cultivated agarwood [38]	Chemical fingerprinting for authenticating origin
Artificial neural networks for oil grading	Use of artificial neural networks to process data on chemical profiles	High accuracy in classification across five oil grades [36,39]	AI-assisted grading for greater objectivity
LESA–MS	Rapid surface authentication	Non-invasive chemical evaluation [32]	Verification of resin authenticity

## 5. Natural Versus Induced Agarwood Production

Agarwood resin forms near wounds or rotting sections of trees in response to stressors such as lightning, fire, insects, or microbes. It rarely develops in young, healthy trees and is scarce even in older ones, with yields of only 1–2%. The resin takes 25–30 years to form naturally in the roots, stems, or branches, which makes it a highly valuable product [13].

The demand for agarwood has outstripped the pace at which agarwood naturally regenerates in forests. The defense mechanism of *Aquilaria* is triggered to stop additional harm in the face of stressors. This results in the formation of secondary metabolites such as sesquiterpenes and 2-(2-phenylmethyl) chromone compounds. These compounds accumulate over the years to form agarwood [40]. Agarwood forms when *Aquilaria* trees respond to fungal infection or injury. To accelerate this slow natural process, artificial methods such as wounding, chemical treatment, and fungal inoculation are used; these methods are more sustainable than wild harvesting [9].

## 6. Artificial Induction Techniques

Agarwood can be artificially produced by inducing resin formation through fungal inoculation (e.g., *Fusarium* sp.) or by physically wounding the *Aquilaria* tree [41]. Fungal induction boosts agarwood yield but risks infection by pathogens. Physical methods (drilling and girdling) rely on the tree’s natural defenses, but the outcomes of these methods are less predictable [42]. Chemical induction carries the risks of pollution, tree damage, and toxicity. A safer alternative is fungal inoculation, which mimics natural infection without harmful effects. In a relevant study, *Aquilaria* trees were infected with *Xylaria* sp. for 2 months to induce agarwood production [43]. After 6 months of inoculation, *Lasiodiplodia theobromae* boosted the production of agarwood in healthy *Aquilaria* trees. Fungal inoculation–induced agarwood yields high-quality essential oils rich in sesquiterpenes and is highly similar to natural agarwood in terms of chemical and antimicrobial properties. The required fungi are easy to cultivate and can produce consistent results in a sustainable manner [35]. *Fusarium solani* is highly effective in inducing high resin yields. *Trichoderma harzianum* is another frequently used species, recognized for its dual role in inducing resin production and suppressing pathogen infection. *Lasiodiplodia theobromae*, commonly associated with naturally infected agarwood trees, induces localized necrosis, which prompts oleoresin secretion. Other genera such as *Penicillium* and *Aspergillus* can also induce resin production, but their effectiveness varies depending on the strain and tree species. These fungi stimulate the tree’s natural defense response, leading to the biosynthesis of aromatic resin components [44].

Agarwood quality depends on the fungal strain used, but endophytic fungi often lead to low yields and thus have limited commercial viability. Nonetheless, the resin resulting from fungal infection is similar to natural resin [45]. Artificial agarwood production is challenging and expensive because induction techniques can kill or severely damage the tree. Plant hormones such as jasmonic acid (JA) and salicylic acid(SA) activate defense pathways, mimicking biotic stress. Ethephon, a synthetic chemical that releases ethylene gas, is widely used in plantations for its ability to efficiently induce a stress response. Other compounds such as hydrogen peroxide, acetic acid, and sulfuric acid induce oxidative or chemical stress to initiate resin production. Mild chemical stressors such as fermented sugar solutions (e.g., molasses mixed with yeast) are sometimes used to increase the effectiveness of fungal inoculation [46].

An ideal solution should balance efficiency, tree health, and high-quality resin output [47]. Non-traditional induction techniques, such as injecting trunks with solvents containing ions or microbial additions or those containing ions but no microbial additions, are often used to produce agarwood [48]. These techniques are easier to use and more accessible than traditional ones but may yield inconsistent outcomes in terms of resin output and quality. Furthermore, their misuse can damage or even kill the trees. Numerous tools and techniques have recently been developed for inducing agarwood production—for example, the whole-tree agarwood inducing technique (Agar-Wit), a biological induction technique (Agar-Bit), and the cultured agarwood kit (CA-kit) [49]. The CA-kit infuses an inducing agent into a wound with the help of an aeration device, thereby combining physical wounding with chemical induction. Although high-quality agarwood is produced using this kit, its environmental and public health implications remain unclear.

Figure 2 depicts the induction process involved in transforming healthy agarwood trees into resinous agarwood through both artificial and conventional means.

## 7. Advances in Resin Induction Techniques

Organic farming, chemical induction, biological inducers, and controlled inoculation are modern methods for producing agarwood [50]. Controlled fungal inoculation—inserting specific fungi into drilled holes in *Aquilaria* trees—reduces resin formation time from decades to just a few years. Chemical inducers such as ethylene and methyl jasmonate are often combined with fungal methods to accelerate resin production [51]. These approaches enable large-scale agarwood production in environment-friendly plantations, guaranteeing a consistent supply while protecting wild populations and meeting the demands of the incense and perfume industries.

Accounting for technological advances, we extend Table 1 to present a more thorough overview and comparison of current induction methods, pharmaceutical studies, and their future evolution in Table 2.

## 8. The State of the Art and Future Prospects

The use of agarwood is most common in East Asia and the Middle East [61]. Recent research on agarwood production techniques has focused on improving sustainability, efficiency, and resin quality. Modern inoculation techniques and mindful forestry management are being incorporated into sustainable planting practices to facilitate large-scale agarwood production without overexploiting native tree populations. Traceability and certification are also becoming crucial to maintaining industry standards on ethics and sustainability. Sustainable agarwood production is imperative to avoid the overharvesting of wild *Aquilaria* species. Biotechnology offers effective solutions by enabling in vitro propagation (e.g., tissue culture) for cultivating plants without damaging natural populations [62]. In the future, synthetic biology and metabolic engineering can be combined to produce resin compounds in microbial systems, reducing the pressure on natural agarwood resources.

The manufacturing process of agarwood is expected to become even more efficient with the use of precision agriculture and AI. AI-powered models (Table 1) may be used to track and forecast the ideal circumstances for resin production, and precision agriculture can guarantee that every tree receives the appropriate care required for optimal resin yield [63]. Agarwood manufacturing may become a sustainable sector if environment-friendly processing techniques and sustainable harvesting technology are developed.

## 9. Chemical Composition and Quality Evaluation

Sesquiterpenes and chromones are the primary chemical components of agarwood. This class of terpenes has the chemical formula C_15_H_24_ and comprises three isoprene units. Sesquiterpenes constitute the majority of the chemical composition of agarwood, which greatly enhances its scent as shown in Table 3 [64]. Structurally, these compounds can be classified as alcohol, ketones, aldehydes, and hydrocarbons on the basis of functional groups present. On the basis of molecular structure, the sesquiterpenes found in agarwood can be classified as follows: candinanes, guaianes, eudesmanes, agarospiranes, agarofurans, and prezizanes [65].

Advanced analytical techniques such as GC–MS have been used to identify sesquiterpenes in agarwood. Among the most well-known sesquiterpenes is α-guaiene; it is the most prevalent sesquiterpene in agarwood. This bicyclic compound has a woody and spicy scent [66]. Another key sesquiterpene is β-caryophyllene, which is known for its analgesic and anti-inflammatory properties. It has a spicy, woody scent. Agarospirol is a characteristic sesquiterpene alcohol that provides agarwood with its distinct, nuanced fragrance [67]. Selinene, found in α and β forms, has been associated with the woody, fresh scent of agarwood [68].

Agarwood also contains chromones at considerable levels. These compounds are produced by a few plant species, such as *Aquilaria* spp., *Eremophila georgei*, and *Bothriochloa ischaemum* (Gramineae) [65]. The 2-(2-phenylethyl) chromone derivatives that agarwood are renowned for have been found in approximately 40 types of agarwood. On the basis of their molecular skeleton, chromones can be divided into 5, 6, 7, 8-tetrahydro-2-(2-phenylethyl) chromones, diepoxy-tetrahydro-2-(2-phenylethyl) chromones, and related chromones [69]. Chromones vary considerably across species. To extract and isolate the monomers in agarwood extracts—often ethanol extracts—researchers studying chromones use various techniques, such as nuclear magnetic resonance and liquid chromatography (LC)–MS, which facilitate the identification of the chromone structures [70].

Agarwood’s scent changes according to its quality. The scent of premium agarwood is usually earthy, sweet, and woody with a rich and long-lasting fragrance. However, low-quality agarwood may have faint or overpowering notes or even an overly pungent smell. Olfactory analysis, a sensory assessment method, can yield a qualitative profile of the fragrance. A small sample of agarwood is burned in traditional methods of evaluation. High-quality agarwood burns slowly and produces rich, fragrant smoke; by contrast, low-quality agarwood produces light smoke with a mild fragrance [68]. The gradual release of aromatic compounds after burning indicates the purity and richness of the resin. The defense system of the tree influences the development of resin, and resin that is left longer to grow has a richer scent. Thus, older agarwood tends to have a stronger, more pleasant scent, which raises its grade [71].

Agarwood’s essential oils are intricate blends of volatile substances, such as monoterpenes and sesquiterpenes, and aromatic substances, such as phenyl ethyl chromones. The composition of essential oils varies depending on plant species, tree age, and extraction technique. Both traditional medicine and perfumery hold these oils in great regard. Steam distillation is the most common technique for extracting essential oils from agarwood; however, other techniques such as solvent extraction and supercritical fluid extraction are also used. GC–MS is commonly used to investigate the essential oil and chemical composition of agarwood. The main ingredients of agarwood essential oils are α-guaiene and β-caryophyllene; these compounds are the main sources of the oil’s scent. Jinkoh-eremol and agarospirol are oxygenated sesquiterpenes that provide premium agarwood oil with its characteristic rich, woody, and deep scent [72]. Notably, 2-(2-Phenylethyl) chromones enhance the aromatic richness of essential oils by conferring a flowery and pleasant scent [73].

Researchers exploring the metabolite profiles of agarwood have indicated metabolites such as 2-(2-phenylethyl) chromones as key aromatic compounds [46]. Techniques such as LC–MS and nuclear magnetic resonance facilitate chromone profiling. Three main chromones have been identified thus far: tetrahydro-2-(2-phenylethyl), epoxy (2-phenylethyl), and diepoxy-(2-phenylethyl) chromones [74]. Chromones and sesquiterpenes markedly contribute to the aromatic and medicinal properties of agarwood. The chromone content in agarwood varies throughout the production process. Higher-quality agarwood is characterized by higher contents of 2-(2-phenylethyl) chromones [75]. The primary compounds in agarwood—sesquiterpenes and chromones—are categorized by their molecular structures and the plant species they originate from (Figure 3). Sesquiterpenes identified from different agarwood species are listed in Table 2 [76].

**Table 3 ijms-26-08468-t003:** Sesquiterpenes are from different agarwood species.

Sesquiterpenes	*A. sinensis*	*A. subintegra*	*A. crassna*	*A. malaccensis*
*β-Agarofurans*	+ [77]	+ [78]	+ [79]	+ [73]
*α* *-Agarofuran*	+ [73]	---	+ [74]	+ [80]
*Dihydro-* *β* *-agarofuran*	+ [73]	+ [78]	+ [81]	+ [73]
*Epoxy-* *β* *-agarofuran*	+ [73]	----	----	+ [82]
*Baimuxinol*	+ [83]	----	----	----
*Isobaimuxinol*	+ [73]	----	----	----
*Dehydrobaimuxinol*	+ [73]	----	----	----
*Baimuxifuranic acid*	+ [84]	----	----	----
*Nor-keto-agarofuran*	+ [73]	+ [78]	+ [78]	+ [85]
*4-Hydroxyl-baimuxinol*	+ [86]	----	-----	----
*Agarospiranes*	+ [73]	+ [78]	+ [78]	+ [73]
*Isoagarospirol*	-----	----	----	+ [73]
*Oxo-agarospirol (baimuxinal)*	+ [87]	+ [78]	+ [78]	+ [87]
*Baimuxinic acid (Bai Mu Xiang acid)*	+ [88]	----	----	----
*Acorenone B*	----	+ [78]	+ [78]	----
*4-epi-15-Hydroxyacorenone*	+ [76]	+ [88,89]	+ [78,81]	----
*4-epi-10-Hydroxyacoronene*	+ [73]	----	----	----
*15-Hydroxyacorenone*	+ [89]	----	----	----
*Eudesmanes*	+ [73]	+ [78]	+ [78]	+ [73]
*(5S,7S,10S)-()-Selina-3,11-dien-9-on*	----	+ [78]	+ [78]	+ [78]
*Agarol (11(13)-eudesmen-12-ol)*	----	----	----	+ [73]
*Selina-3,11-dien-14-ol*	----	+ [78]	+ [78]	----
*Isolongifolene*	+ [90]	----	----	----
*α-Eudesmol*	+ [73]	----	----	----
*α-Copaen-11-ol*	+ [73]	----	----	----
*β* *-Eudesmol*	+ [73]	+ [78]	+ [78]	+ [78]
*g-Selinene*	+ + [90]	----	----	----
*d-Selinene*	+ [90]	----	----	----
*α-Copaene-8-ol 43*	+ [73]	----	----	----
*β-Maaliene 36*	+ [80]	----	----	----
*β-Eudesmol acetate*	----	+ [78]	+ [78]	----
*α-Selinene*	+ [91]	----	+ [91,92]	----
*Eudesm-7(11)-en-4a-ol*	+ [91]	----	----	----
*Eremophilanes*	+ [73]	----	----	+ [73]
*(+ )-(4S,5R)-Dihydrokaranone*	----	+ [78]	+ [78]	+ [93]
*Dehydro-jinkoh-eremol*	----	----	----	+ [94]
*Calarene*	+ [73]	----	----	----
*7b-H-9(10)-ene-11,12- poxy-8-oxoeremophilane*	+ [73]	----	----	----
*11-Hydroxy-valenc-1(10)-en-2-one*	+ [7]	----	----	----
*Ligudicin C*	+ [95]	----	----	----
*(+)-11-Hydroxyvalenc-1(10),8-dien-2-one*	+ [73]	----	+ [96]	----
*Valencene*	+ [73]	+ [78]	+ [78]	+ [73]
*Nootkatone*	+ [73]	----	----	----
*α-Guaianes*	----	+ [78]	+ [78]	+ [78]
*α-Bulnesene*	----	+ [78]	+ [78]	+ [97]
*(-)-Epoxyguai-11-ene (epoxybulnesene)*	----	+ [78]	+ [78]	+ [78]
*(-)-Guaia-1(10),11-dien-15-ol*	----	+ [78]	+ [78]	+ [78]
*(-)-Guaia-1(10),11-dien-15-al*	----	----	----	+ [73]
*(-)-Guaia-1(10),11-diene-15-carboxylic acid*	----	----	----	+ [73]
*Rotundone*	----	----	----	+ [73]
*Sinenofuranol*	+ [73]	----	----	----
*Sinenofuranal*	+ [73]	----	----	----
*Viridifloorol*	+ [73]	----	----	----
*Ledol*	+ [73]	----	----	----
*Longifolene*	+ [73]	----	----	----
*Aromadendrene*	----	----	+ [87]	----
*Guaiol*	+ [90]	----	----	----
*Chamaejasmone*	----	----	----	+ [90]
*Chamaejasmone*	----	----	----	+ [90]
*Candinanes*	----	----	----	+ [98]
*Gmelofuran*	----	----	----	+ [98]
*(7b,8b,9b)-8,9-Epoxycalamenen-10-one*	----	----	+ [92]	----
*Prezizanes*	----	----	----	+ l [91]
*Jinkohol*	----	----	----	+ [76]
*Daphnauranol B*	----	----	----	+ [73]
*Daphnauranol C*	----	----	----	+ [92]
*Daphnauranol D*	----	----	----	+ [92]
Others				
*Patchoulialcohol*	+ [15]	----	----	----
*(+)-8b-Hydroxy-longicamphenylone*	+ [99]	----	----	----
*Valerenol*	----	----	+ [78]	----
*Valerenic acid*	+ [73]	----	+ [78]	----
*Valerenal*	+ [99]	----	+ [96]	----
*Dihydro-neoclovene*	----	----	+ [96]	----
*2,6-Dimethyl-10-methylene-12-oxatricyclo tridec-2-ene 2*	+ [73,91]	----	-----	----
*β-Elemene*	----	+ [78]	+ [78]	----
*α-Bisabolol acetate*	----	+ [78]	----	----
*α-Caryophyllene*	+ [73]	----	----	----
*α-Humulene*	+ [73]	----	----	----
*Humulene diepoxide A*	+ [100]	----	----	----
*Kobusone*	+ [73]	----	----	----
*Santalol*	+ [91]	----	----	----
*(E)-Nerolidol*	----	+ [78]	+ [78]	+ [78]
*Caryophyllenol-II*	+ [73]	----	----	----
*Caryophylleneoxide*	+ [73]	----	----	+ [73]
*Baldrina*	+ [99]	----	----	----
*α-Muurolene*	+ [99]	----	----	----
*Elemol*	+ [97]	----	----	----
*cis-Z-α-Bisabolene epoxide*	+ [98]	----	----	----
*Cubenol*	+ [91]	----	----	----
*1,2,5,5,8 α-Pentamethyl-1,2,3,5,6,7,8,8 α -octahydronaphthalen-1-ol*	+ [91]	----	----	----
*1,5,9-Trimethyl-1,5,9-cyclododecatriene*	+ [73]	----	----	----
*Aquilanol A*	----	----	----	+ [73]
*Aquilanol B*	----	----	----	+ [73]
*12-Hydroxyhumula-2Z,6E,9E-triene*	----	----	----	+ [73]
*14-Hydroxy-α-humulene*	+ [89]	----	----	----

## 10. Exploration of Molecular Mechanisms Underlying Agarwood Production

Effective agarwood manufacturing requires an understanding of the molecular mechanisms underlying the secondary metabolite biosynthetic pathways involved in resin production. Plant metabolomics has been extensively used to assess dynamic changes in metabolites across species, tissues, and developmental stages. In *Rhodiola imbricata*, RNA-seq-based molecular bioprospecting techniques are used to identify regulatory genes and explain how variations in biosynthetic gene expression profiles influence metabolite content [101]. Combined analysis of transcriptomic and metabolic data may reveal functional genes as well as primary and secondary metabolic pathways in plants [13,102].

Metabolomic analyses are performed to identify key secondary metabolites essential for the distinct scent of agarwood—for example, chromones and sesquiterpenes. These analyses unveil the biosynthetic pathways induced by stressors such as fungal infection and mechanical damage, as illustrated in Figure 4. Stress signals such as pathogen attack or mechanical injury are perceived by plant receptors, activating downstream signaling cascades. NADPH oxidase triggers the production of hydrogen peroxide (H_2_O_2_), while calcium influx via Ca^2+^–calmodulin complexes further amplifies defense responses. These signals stimulate reactive oxygen species (ROS) accumulation, leading to salicylic acid (SA) and jasmonic acid (JA) pathway activation, as well as programmed cell death (PCD). The interplay of JA, SA, and ethylene (ET) signaling regulates transcription factors such as WRKY, MYB, and MYC, which bind to cis-regulatory elements of terpene biosynthesis genes. Activation of mevalonate (MVA) and methylerythritol phosphate (MEP) pathways plays a central role in sesquiterpene biosynthesis, involving key enzymes such as 3-hydroxy-3-methylglutaryl-coenzyme A reductase (HMGR) and **1**-deoxy-D-xylulose-5-phosphate synthase (DXS) [13]. Studies have shown that sesquiterpene synthase genes like *ASS1* and *ASS2* are upregulated in response to mechanical wounding and fungal infection, which are commonly used methods to induce agarwood formation [103]. Additionally, transcription factors such as WRKY, MYC2, and bHLH are known to regulate gene expression under stress conditions that lead to resin accumulation. These responses are often mediated by signal transduction pathways involving JA and SA, which activate genes involved in secondary metabolite biosynthesis [104,105].

A study [102] performed transcriptomic and metabolic analyses of six tissue types (root, stem, leaf, seed, husk, and callus) from *A. sinensis* to evaluate their potential as alternatives in agarwood cultivation. GC–MS performed in the aforementioned study revealed 331 distinct metabolites, with terpenoids being the most prevalent group (22.89%). Sesquiterpenes constituted 51% of the total terpenoid content. Notably, most sesquiterpenes were detected in callus tissue [105]. The biosynthesis of sesquiterpenes and terpenoids from agarwood leaf–derived callus tissue is illustrated in Figure 5. This figure depicts the process from leaf to callus induction and presents the major classes of compounds. The figure was prepared on the basis of original experimental data from our laboratory. The highest variability in metabolite content was observed in the husk and callus tissues. RNA-seq data production genes involved in sesquiterpene production were markedly enriched, particularly in the callus tissue. The aforementioned study revealed similar transcriptional profiles between the roots and stems and highlighted a predominance of sesquiterpene synthase genes in callus tissue (60%). These results offer valuable insights into sesquiterpene production and may guide the search for alternative agarwood sources [106].

The integrated approach opens new avenues for synthetic biology applications and sustainable production techniques, which may reduce the reliance on natural agarwood sources and thus support conservation efforts.

## 11. Pharmacological Properties

Agarwood essential oil has antibacterial, anti-inflammatory, antioxidative, anti-asthmatic, anti-convulsive, and analgesic properties, which make it valuable for use in pharmaceutical as mentioned in Table 4. Evidence suggests that agarwood essential oil holds promise in the treatment of prostate, pancreas, breast, and colorectal cancers [107]. Agarwood was demonstrated to exert hepatoprotective effects in a mouse model of chemically induced liver injury [107].

Most pharmacological applications of agarwood remain confined to traditional medicine systems, particularly Traditional Chinese Medicine (TCM), Ayurveda, and Unani [18]. Within these systems, agarwood has been used for centuries to manage a variety of ailments, including asthma, digestive disorders, pain, and anxiety.

To date, no agarwood-based pharmaceuticals have been approved as registered drugs by major international regulatory agencies such as the U.S. Food and Drug Administration (FDA) or the European Medicines Agency (EMA). However, in several Asian countries—notably China, Japan, Korea, and parts of Southeast Asia—agarwood is used as an active ingredient in registered herbal formulations or licensed traditional medicine preparations [108].

For example, in China, agarwood (沉香, *Chen Xiang*) is officially listed in the Chinese Pharmacopoeia [109] and is included in formulations for treating gastrointestinal conditions and inducing sedation. In Japan, agarwood is used in Jinkō-To aimed at treating digestive discomfort and qi stagnation. Similarly, in Malaysia and Indonesia, agarwood extracts are available as over-the-counter herbal supplements and essential oils, regulated by local traditional medicine authorities [110]. A major limitation on the medical application of agarwood lies in its high cost and limited availability. Agarwood resin forms slowly and unpredictably, and even with artificial induction, yields remain low. These factors hinder large-scale production, standardization, and cost-effective development for modern medicine.

**Table 4 ijms-26-08468-t004:** Agarwood’s bioactive compounds and their potential therapeutic applications.

Property	*Aquilaria* Species	Key Findings	Mechanism/Active Compounds	Potential Applications
Sedative and anxiolytic	*A.* *malaccensis* *A. sinensis*	Modulates central nervous system activity and reduces stress and insomnia	GABAergic interaction; sesquiterpenes (agarospirol and jinkoh-eremol)	Anxiety and sleep disorders
Analgesic and anti-inflammatory	*A.* *crassna* *A.* *malaccensis*	Alleviates pain and inflammation in animal models	Inhibition of tumor necrosis factor-α and interleukin-6 [111]; various compounds (linalool, 10-epi-γ-eudesmol, and agarospirol) [112,113]	Arthritis [114] and inflammatory conditions
Antioxidative	*A.* *sinensis* *A. agallocha*	Scavenge free radicals [115] and prevents oxidative damage	Phenolic compounds and DPPH radical reduction	Oxidative stress-related diseases
Antimicrobial	*A.* *agallocha*	Effective against bacteria (e.g., *Staphylococcus aureus* and *Escherichia coli*) and fungi (e.g., *Candida albicans*) [116]	Baimuxinol, agarospirol, and essential oils	Bacterial/fungal infections [117]
Antidiabetic	*A.* *malaccensis*	Reduces blood glucose level and improves insulin sensitivity [118]	Iriflophenone 3-C-β-glucoside and ethanol/methanol extracts	Diabetes management [119]
Anticancer	*A.* *crassna*	Inhibits angiogenesis and induces apoptosis in cancer cells	β-caryophyllene [120] and phenanthrene derivatives	Adjunct cancer therapy [27]

Table 3 presents key bioactive compounds present in agarwood and their potential therapeutic applications. Preclinical studies have *indicated* various pharmacological benefits of *Aquilaria*. The diversity of bioactive compounds in agarwood highlights its potential for exerting multitarget effects. In the future, well-designed pharmacokinetic studies should be conducted to investigate dose–response relationships. Moreover, human trials should be conducted to validate the safety and efficacy of agarwood’s bioactive compounds.

## 12. Agarwood Extract: Variability, Contaminants, and Regulatory Challenges

Agarwood extract is a highly valuable ingredient in traditional medicine, fragrance, and incense. It is extracted from the resinous wood of the *Aquilaria* spp. The chemical composition of agarwood extract, as well as corresponding biological activity and possible toxicity, varies depending on the plant species, agarwood origin, and extraction technique. Agarwood contains phenolic chemicals, several sesquiterpenes, and other bioactive compounds that can be toxicologically hazardous when ingested in high amounts [121,122]. The risk of adulteration cannot be ignored because unlicensed harvesting and agarwood processing may lead to contamination by heavy metals, synthetic compounds, or other plant elements. The lack of international standards limits efforts to ensure the safety of agarwood extracts. Despite centuries of traditional use, few solutions for the aforementioned problems are available. Thus, agarwood products should be used sparingly, particularly when consumed or administered topically. Agarwood consumption should not be a substitute for consultation with a health-care professional, especially if the individual has a medical history or is on other drugs [123]. 

## 13. Key Insights from Major Studies

Agarwood formation is a complex biochemical process induced by biotic and abiotic stressors, which activate the secondary metabolism of the plant and lead to the production of key aromatic compounds such as sesquiterpenes and 2-(2-phenylethyl) chromones. Despite advances in artificial induction techniques—for example, physical wounding, fungal inoculation, and chemical elicitation—the quality, yield, and consistency of resin production remain variable and highly dependent on genetic, environmental, and methodological factors. Tools such as the CA-kit and Agar-Bit are major steps forward, but their long-term ecological impact requires further assessment.

Ensuring the authenticity and traceability of agarwood products is another critical challenge. The lack of standardized grading systems and the prevalence of synthetic or adulterated products have compromised consumer trust and fair-trade practices. DNA barcoding and molecular identification methods—focusing on the *matK*, *rbcL*, internal transcribed spacer, and trnL-trnF regions—have emerged as effective tools for species authentication. These techniques, particularly when combined with chemical profiling and metabolomics, offer promising solutions for quality control, regulatory compliance, and conservation efforts.

Chemical analysis through GC–MS and LC–MS has accurately identified bioactive compounds, supporting quality assessment and pharmacological exploration. Sesquiterpenes and chromones not only define the aromatic profile of agarwood but also exhibit potent antibacterial, antioxidative, anti-inflammatory, analgesic, hepatoprotective, and anticancer activities. These pharmacological attributes underscore agarwood’s therapeutic potential and justify its inclusion in both traditional and modern pharmacopeias.

Recent advances in transcriptomic and metabolomic studies have deepened our understanding of the molecular mechanisms underlying resin biosynthesis, particularly the roles of sesquiterpene synthase genes and stress-related pathways. These insights may inform targeted genetic and biotechnological strategies for enhancing resin production in a sustainable manner, thereby expanding agarwood’s availability while reducing pressure on wild plants.

Because of overharvesting, illegal trade, and habitat loss, sustainability is a major concern in agarwood production. This challenge may be overcome by adopting in situ and ex situ conservation, plantation-based cultivation, and agroforestry, which would reduce pressure on wild plant populations. Biotechnological tools, such as microbial or chemical resin induction, tissue culture for mass propagation, and genetic profiling, can enhance yield while protecting biodiversity. Regulatory support through CITES and traceability methods such as DNA barcoding can further promote legal and ethical trade. Together, these strategies can balance the increasing global demand for agarwood with a commitment to long-term environmental sustainability.

## 14. Conclusions

The future of agarwood production lies at the intersection of sustainable cultivation, molecular authentication, and innovative biological techniques. Continued interdisciplinary research integrating plant biology, chemistry, genomics, and environmental science is essential for optimizing agarwood production, ensuring ethical trade, preserving endangered *Aquilaria* sp., and realizing the medicinal and commercial potential of this precious natural resources.

## Figures and Tables

**Figure 1 ijms-26-08468-f001:**
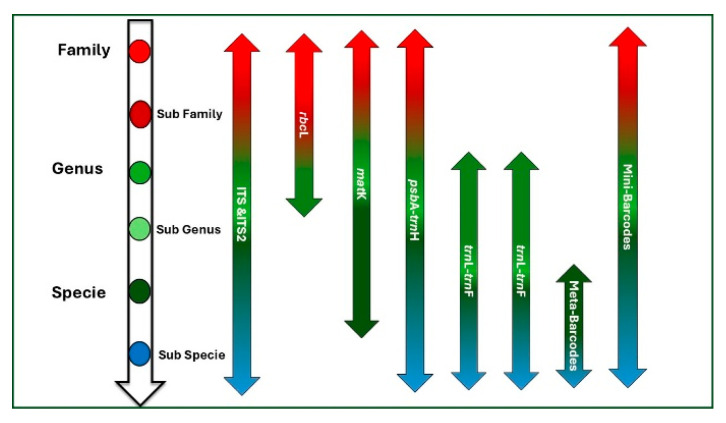
Taxonomic resolution of common plant DNA barcodes from family to subspecies level. Figure inspired by [20] and created by the authors.

**Figure 2 ijms-26-08468-f002:**
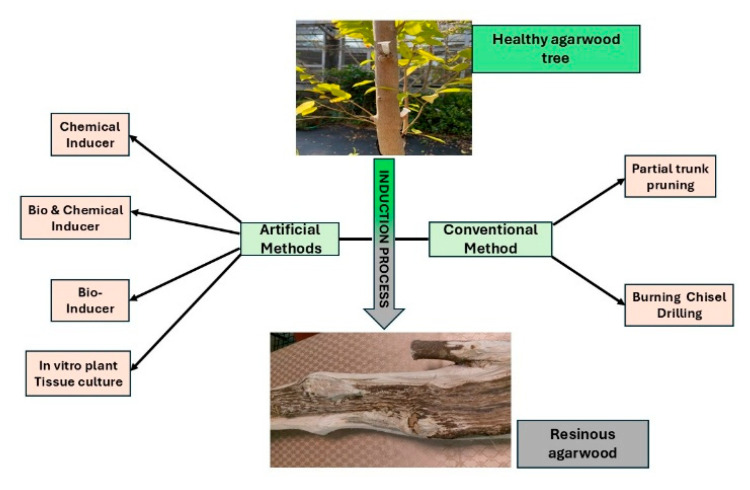
Schematic overview of artificial and conventional methods used for agarwood resin induction.

**Figure 3 ijms-26-08468-f003:**
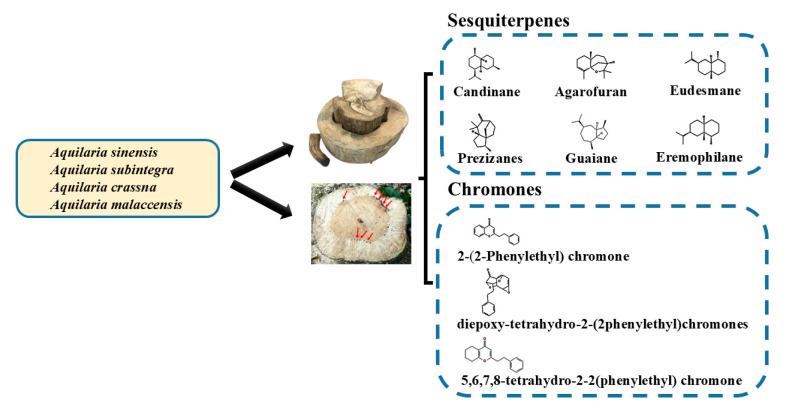
Primary compounds in agarwood (sesquiterpenes and chromones) and their molecular skeletons.

**Figure 4 ijms-26-08468-f004:**
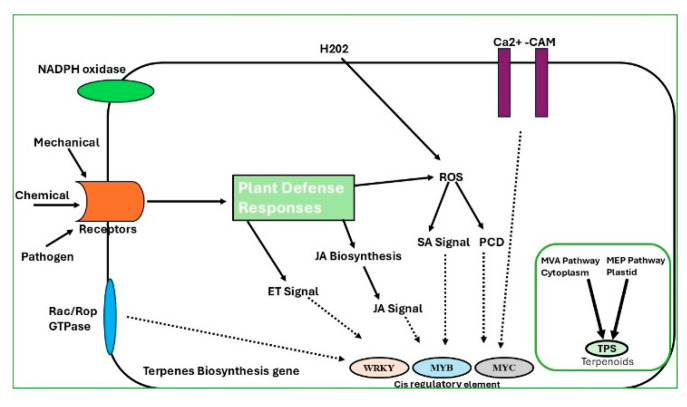
Overview of plant defense signaling pathways regulating terpene biosynthesis. Pathogen or mechanical stimuli activate receptors, triggering ROS, JA, SA, and ET signaling. Transcription factors (WRKY, MYB, and MYC) regulate terpene biosynthetic genes via MVA and MEP pathways, leading to terpenoid production.

**Figure 5 ijms-26-08468-f005:**
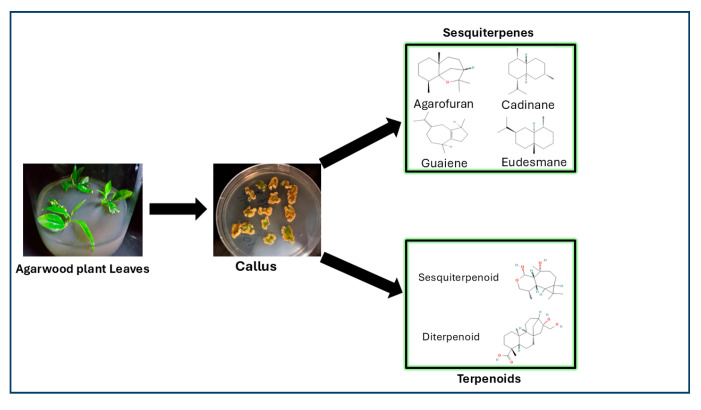
Production of sesquiterpenes and terpenoids from agarwood callus.

**Table 2 ijms-26-08468-t002:** Fungal induction versus physical induction.

Criteria	Fungal Induction	Physical Induction (Artificial Induction)
Mechanism	Introduction of specific fungi (e.g., *Fusarium* sp.) to induce resin production [44].	Physical wounding (e.g., drilling and girdling) to stimulate the tree’s natural defense mechanism and thus resin production [52].
Time required for resin production	In general, the production time is short because of the direct action of the fungi on the wood [53].	The production time may be prolonged because these techniques rely on the tree’s natural response to physical damage.
Resin quality	High-quality resin with desirable chemical composition, often influenced by the fungal strain used [54].	Quality can vary widely depending on the extent of wounding and the health of the tree [55].
Control and consistency	More control over the quality and quantity of resin produced, with consistent results [51].	Less consistent because the outcome depends on the tree’s response to physical injury [6].
Environmental impact	Risk of unintended spread of pathogens [56].	Minimal direct environmental impact but overuse can lead to tree damage or death [57].
Cost	Moderate to high because specialized fungal cultures and careful application are required [11].	Relatively low because manual or mechanical wounding is used; however, cost can increase if extensive labor is required [58].
Sustainability	More sustainable because the process can be carefully controlled and optimized; however, these techniques are dependent on appropriate management of fungi [45].	Sustainable if managed appropriately; however, overuse can lead to overharvesting and depleting tree resources [59].
Potential risks	Pathogenic fungi may spread to non-target trees in the case of improper management [60].	The tree may die in the case of excessive or improper wounding [54].

## Data Availability

All necessary data generated or analyzed during this study are included in this article.

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
