# Peer review of "Agarwood in the Modern Era: Integrating Biotechnology and Pharmacology for Sustainable Use"

_ijms, 2025, doi:10.3390/ijms26178468_

Round 1

Reviewer 1 Report

Comments and Suggestions for Authors

The review paper is interesting however, the manuscript needs a word by word and sentence by sentence English proofreading by a native English speaker to enhance scientific language and correct minor typos. Also, add two more tables in the study. Revise the Figure 1 and add some pictures such as resins of Aquilaria and Gyrinops tree and wood, etc. Also add some more pictures about the Agarwood extracts and color scheme, etc, A review paper must have a detail pictorial over for future readers to clearly understand the agarwood species, difference between their genus. etc. I would also like to ask is Aquilaria and Gyrinops are different? what are the difference between them. Which part of these are used for Starwood extraction. how many methods and what is the procedure of agarwood extraction? etc. chemical structure and the other biosynthesis related materials need to be added. 

Comments on the Quality of English Language

The manuscript need to be thoroughly revised and approved by a native English speaker. 

Author Response

The review paper is interesting however; the manuscript needs a word by word and sentence by sentence English proofreading by a native English speaker to enhance scientific language and correct minor typos. Also,
1) Add two more tables in the study.
Ans: Tables have been added.
2) Revise Figure 1
Ans: Figure 1 is revised and now is the new Figure 2.
3) Add some pictures such as resins of Aquilaria
Ans: Four new pictures have been added.
4) Gyrinops tree and wood
Ans: Gyrinops tree and wood are not the main target of this study. We are focused on agarwood ; therefore, we removed the Gyrinops.
5) Also add some more pictures about the Agarwood extracts and color scheme, etc.,
Ans: We added some new pictures like Figure 1, Figure 3 and Figure 4.
6) A review paper must have a detail pictorial over for future readers to clearly understand the agarwood species, and the difference between their genera. etc.
Ans: Agarwood species and their genus explained via Table 3.
7) I would also like to ask is Aquilaria and Gyrinops are different?
Ans: Gyrinops tree and wood are not the main target of this study. We are focused on agarwood ; therefore, we removed the Gyrinops.
What is the difference between them. Which part of these are used for Starwood extraction. how many methods and what is the procedure of agarwood extraction? ( agarwood extraction procedure in explained etc. chemical structure and the other biosynthesis related materials need to be added.
Ans: We appreciate this comment; We explained induction techniques of agarwood including artificial and natural induction from line 189 to 248 . Figure 2 to also elaborate the induction processes. In Part chemical composition and quality evaluation , compounds and methods are narrated about the production sesquiterpene and terpenoids. In figure 4 the biosynthesis mechanism, regulatory elements and the gene involved in the response are explained.

Reviewer 2 Report

Comments and Suggestions for Authors

While reading the manuscript ijms-3731879, I developed a negative attitude towards the presented text. The authors often repeat information about the chemical composition of agarwood, its commercial value, its rarity in nature, the need for control, and other things. At the same time, there is no information important for understanding the problem - what is "agarwood"? Under this term, the authors present one thing (plants) and then another (wood), without any explanation. For this reason, the text of the manuscript is very difficult to understand. For example, (line 41) "Agarwood, a plant belonging to the Thymelaeaceae family", but in the next three paragraphs (line 52) "healthy Aquilaria trees can generate a limited quantity of agarwood", (lines 55-56) "Agarwood forms in Thymelaeaceae trees due to microbial infection" and (lines 65-66) "Healthy Aquilaria trees don't produce agarwood". What is this review about, plants or wood?

Another major drawback of the review is that the authors list two plant genera (Aquilaria and Gyrinops) as agarwood, but the entire review is limited to Aquilaria species.

The authors pay a lot of attention to "species authentication and DNA barcoding technology" but do not explain what scientific or commercial value these studies represent. In this regard, the relevance of this review is extremely low.

Data on the phytochemical composition and pharmacological value of agarwood have previously been presented in other reviews since 2020; for example, Chen et al., 2022. In turn, the presentation of data in this review seems to me to be not very successful - the principle of presenting information should be changed for Table 2.

The authors indicated a description of the innovative induction techniques in the objectives of the review, but the text of the manuscript does not contain any detailed discussions of this aspect.

Author Response

While reading the manuscript ijms-3731879, I developed a negative attitude towards the presented text. The authors often repeat information about the chemical composition of agarwood, its commercial value, its rarity in nature, the need for control, and other things. At the same time, there is no information important for understanding the problem –
what is "agarwood"?
Ans: Answer is given in lines 41-43.
Under this term, the authors present one thing (plants) and then another (wood), without any explanation. For this reason, the text of the manuscript is very difficult to understand. For example, (line 41) "Agarwood, a plant belonging to the Thymelaeaceae family", but in the next three paragraphs (line 52) "healthy Aquilaria trees can generate a limited quantity of agarwood", (lines 55-56) "Agarwood forms in Thymelaeaceae trees due to microbial infection" and (lines 65-66) "Healthy Aquilaria trees don't produce agarwood". What is this review about, plants or wood?
Ans: The review is about agarwood. There were some typing errors. They are proofread and corrected from line 41 to 66.
Another major drawback of the review is that the authors list two plant genera (Aquilaria and Gyrinops) as agarwood, but the entire review is limited to Aquilaria species.
Ans: The main target is Aquilaria. Gyrinops aren’t main focus of article, so therefore we rewrite or remove typos and errors.
The authors pay a lot of attention to "species authentication and DNA barcoding technology" but do not explain what scientific or commercial value these studies represent. In this regard, the relevance of this review is extremely low.
Ans: We really appreciate this comment, so in this context we add a new Figure and some information to clear the idea of DNA barcoding (Figure 1).
Data on the phytochemical composition and pharmacological value of agarwood have previously been presented in other reviews since 2020; for example, Chen et al., 2022. In turn, the presentation of data in this review seems to me to be not very successful - the principle of presenting information should be changed for Table 2.
Ans: We added some more innovative information and also added 1 more table (Table 1) to clear up the new techniques and their findings with main key factors. We hope it will clear up the suggestion.
The authors indicated a description of the innovative induction techniques in the objectives of the review, but the text of the manuscript does not contain any detailed discussions of this aspect.
Ans: Induction techniques are now written in detail to clear the suggestion with Figure illustration (Figure 2)

Reviewer 3 Report

Comments and Suggestions for Authors

This manuscript attempts to review current knowledge on agarwood production, authentication, and pharmacological properties with a focus on sustainability. While the topic is timely and of commercial relevance, the manuscript suffers from major structural, conceptual, and scientific deficiencies.
Many claims in the abstract, such as the use of genetic modification, artificial intelligence, tissue culture, and biotechnology, are not supported by content in the main text. 
The review predominantly catalogs facts without critical evaluation of methods, unresolved questions, or conflicting evidence. Several sections, including those on pharmacology and toxicity, are poorly referenced, purely descriptive, or redundant, lacking critical analysis or depth. Multiple paragraphs repeat the same ideas, especially in sections describing induction methods and chemical constituents. The section titled “Molecular mechanisms” does not present actual mechanisms, and the concluding section fails to synthesize the review or offer meaningful insights.
In its current form, the manuscript does not meet the standards for publication. 

Specific comments:
Title: The title is misleading. Although it promises integration of biotechnology and pharmacology for sustainable use, the manuscript contains no substantive discussion of biotechnological approaches, and its treatment of sustainability is cursory. The scope implied by the title is not matched by the actual content.
Abstract: 1. The statement regarding the application of artificial intelligence and machine learning in agarwood identification is not substantiated anywhere in the main text. No methods, examples, or references are provided to explain how AI enhances species identification or phytochemical analysis. This appears to be an unfounded claim and should either be removed from the abstract or supported with appropriate content and references in the manuscript. 2. The claim that "genetic modification" is among the advanced induction methods improving yield and quality is not supported in the main text. The manuscript does not provide any discussion or reference to actual genetic modification work in Aquilaria spp. This statement is misleading and should be either substantiated with content and citations or removed from the abstract. 3. The statement that “tissue culture facilitates large-scale production” is unsubstantiated in the manuscript. Tissue culture is mentioned only once and is not elaborated upon or supported by any experimental data, literature review, or methodological detail. This claim should either be developed with references and examples or removed from the abstract to avoid misleading readers.
Line 41 and further throughout the text: “Agarwood, a plant belonging to the Thymelaeaceae family…” This is a fundamental taxonomic error. Agarwood is not a plant, but rather a resinous product that forms in response to biotic or abiotic stress within certain trees of the genus Aquilaria and Gyrinops, which are members of the family Thymelaeaceae. Throughout the manuscript, the authors frequently use “agarwood” as if it were synonymous with the trees that produce it.
Lines 103–131: The explanation of DNA barcoding is overly simplistic and largely descriptive. While some commonly used loci are named, there is no critical discussion of their strengths, weaknesses, or application limits in Aquilaria authentication. The section lacks depth and does not reflect current advances in molecular diagnostics.
Line 121: The sentence "Notable species include Aquilaria malaccensis and A. crassna" is fragmentary and uninformative. It lacks context, justification, or connection to the cited reference.
Line 122: The phrase “Taxon DNA is used to calculate species identification rates” is vague and non-standard. It is unclear whether this refers to a software package or a general method. 
Lines 157–185: This section lacks novelty and primarily repeats earlier content. It misses the opportunity to discuss modern techniques in product authentication and their role in regulatory oversight. The narrative should be expanded to include practical tools used in commercial or forensic settings.
Lines 200-245: The section lacks scientific rigor. While methods are mentioned (e.g., fungal or chemical induction), the authors provide no comparative data, mechanisms, or performance metrics. The discussion remains at a descriptive level without evaluating the effectiveness or limitations of each approach. Terms like "chemical inducers" or "biological agents" are used broadly, but no specific compounds or strains are named or characterized.
Lines 250–259: The authors appear to equate the use of fungal inoculation with "biotechnology advancements," but this is conceptually incorrect. Fungal inoculation, unless coupled with molecular tools or genetic engineering, is a biological method—not a biotechnological one. 
Lines 263–283: The “Present Advancements and Prospects for the Future” section is speculative and lacks depth. It makes broad claims about AI, genetic engineering, and sustainability without providing supporting references or technical detail. Moreover, it does not synthesize earlier content or offer a concrete roadmap for future research. For example, the statement about “AI-powered models” predicting optimal conditions for agarwood resin production is speculative and cited inappropriately. The referenced study (Zhu et al., 2020) concerns 3D-printed multifunctional materials and is unrelated to the topic.
Lines 363–403 (“Molecular mechanisms” section): This section is misleadingly titled. It does not present or review molecular mechanisms underlying agarwood formation in any meaningful way. Instead, it summarizes general metabolomics and transcriptomic analyses without identifying specific genes, pathways, or regulatory systems involved in resin biosynthesis. The content is superficial and tangential to the article’s focus and lacks integration with the broader narrative on induction methods or sustainable use.
Figure 2: The figure is not related to molecular mechanisms as the title suggests. Instead, it presents a generic schematic of induction triggers (e.g., stress, fungi), which merely repeats earlier content. It lacks any molecular detail (e.g., gene names, biosynthetic steps) and does not support the section’s title or claims. Its inclusion in this section is redundant and scientifically uninformative.
Lines 408–444: The section on medicinal properties and toxicities is overly descriptive and lacks adequate referencing. Claimed biological activities are not supported by experimental data, citations, or mechanistic discussion. The toxicity content is similarly superficial. A critical scientific review should provide detailed references, specify experimental contexts, and distinguish between traditional use and evidence-based pharmacology.
Lines 417–424: The “Pharmacological Properties” section lacks any analytical content. It merely lists proposed therapeutic effects in a table without interpretation or narrative discussion. The authors should synthesize the data, evaluate the quality and limitations of the evidence, and provide critical insight. A table without context is insufficient for a scientific review.
Lines 425–427: The hepatoprotective property is arbitrarily separated into a standalone section titled “Other Pharmacological Effects” without justification. This content should be integrated into the main pharmacological discussion or table for consistency and clarity. As it stands, the structure appears disjointed and editorially unbalanced.
Lines 430–444: The section titled “Safety Issues and Possible Toxicities” does not fulfill the expectations of the title. It lacks toxicological data, safety assessments, or regulatory discussion. Instead, it repeats general observations from previous sections without providing new or specific information. The title should be revised to reflect the actual content, or the section should be significantly expanded and properly referenced to address toxicity and safety.
Lines 445–472: The manuscript lacks a proper conclusion. The final section, titled “Summarized insights from selected key studies,” repeats technical content from earlier sections rather than synthesizing the main findings or offering critical closing thoughts. A scientifically structured review requires a clear, concise conclusion that integrates the key themes, reflects on limitations, and proposes future research directions.

Comments on the Quality of English Language

The manuscript contains frequent grammatical and punctuation errors, awkward phrasing, and unclear sentence structures, all of which impair readability. Thorough language editing by a fluent English speaker or professional service is essential prior to publication.

Author Response

This manuscript attempts to review current knowledge on agarwood production, authentication, and pharmacological properties with a focus on sustainability. While the topic is timely and of commercial relevance, the manuscript suffers from major structural, conceptual, and scientific deficiencies.

Many claims in the abstract, such as the use of genetic modification, artificial intelligence, tissue culture, and biotechnology, are not supported by content in the main text.

Ans: We appreciate this comment in this context we add some figures like Figure 1 about DNA barcoding, Table 1 in which new technology and their key findings are presented to summarize the idea. Also, we add Figure 3 and Figure 4 to clear the whole idea. We hope that are more than enough.

The review predominantly catalogs facts without critical evaluation of methods, unresolved questions, or conflicting evidence. Several sections, including those on pharmacology and toxicity, are poorly referenced, purely descriptive, or redundant, lacking critical analysis or depth. Multiple paragraphs repeat the same ideas, especially in sections describing induction methods and chemical constituents.

Ans: The paragraphs in induction methods are proofreads, multiple sentences and paragraphs are merged to clear the idea 189 to 251 . Also, the section pharmacology sections are merged into single headings 425 t0 443 including Table 3. Some citations are revised. Typos errors and repetition of paragraphs and sentences are corrected.

The section titled “Molecular mechanisms” does not present actual mechanisms, and the concluding section fails to synthesize the review or offer meaningful insights.

Ans: Molecular mechanism was explained in Figure 4 to clarify the mechanism also highlights this part in article from line 385 to 387.

Specific comments: Title: The title is misleading. Although it promises integration of biotechnology and pharmacology for sustainable use, the manuscript contains no substantive discussion of biotechnological approaches, and its treatment of sustainability is cursory. The scope implied by the title is not matched by the actual content.

Ans: we appreciate that comment so in this context we added some information about sustainability and its future goal from line from 278-282 and also we added a paragraph in conclusion from line 501to 508

Abstract: 1. The statement regarding the application of artificial intelligence and machine learning in agarwood identification is not substantiated anywhere in the main text. No methods, examples, or references are provided to explain how AI enhances species identification or phytochemical analysis. This appears to be an unfounded claim and should either be removed from the abstract or supported with appropriate content and references in the manuscript.

Ans: This part is added and cleared now from line 173 to 181 and Table 1 added in manuscript in which new evidence and there results are mentioned with reference. Also, the references are added. (Hipiny et al., 2024)

  1. 2. The claim that "genetic modification" is among the advanced induction methods improving yield and quality is not supported in the main text.

Ans: Genetic modification text is removed.

The manuscript does not provide any discussion or reference to actual genetic modification work in Aquilaria spp. This statement is misleading and should be either substantiated with content and citations or removed from the abstract.

Ans: This part is removed.

  1. 3. The statement that “tissue culture facilitates large-scale production” is unsubstantiated

in the manuscript. Tissue culture is mentioned only once and is not elaborated upon or supported by any experimental data, literature review, or methodological detail. This claim should either be developed with references and examples or removed from the abstract to avoid misleading readers.

Ans: Some important information was added and typos removed, about the tissue culture and callus. We added some information from our lab data and new research from line 401-415.Also the figure five narrates the basic idea of callus.

Line 41 and further throughout the text: “Agarwood, a plant belonging to the Thymelaeaceae family…” This is a fundamental taxonomic error. Agarwood is not a plant, but rather a resinous product that forms in response to biotic or abiotic stress within certain trees of the genus Aquilaria and Gyrinops, which are members of the family Thymelaeaceae. Throughout the manuscript, the authors frequently use “agarwood” as if it were synonymous with the trees that produce it.

Ans: Aquilaria and Gyrinops are two different plants within same family. The manuscript is mainly focused on Aquilaria only, so we removed the Gyrinops from the article.

Lines 103–131: The explanation of DNA barcoding is overly simplistic and largely descriptive. While some commonly used loci are named, there is no critical discussion of their strengths, weaknesses, or application limits in Aquilaria authentication. The section lacks depth and does not reflect current advances in molecular diagnostics.

Ans: DNA barcoding is explained through Figure 1 to clear this suggestion of DNA barcoding.

Line 121: The sentence "Notable species include Aquilaria malaccensis and A. crassna" is fragmentary and uninformative. It lacks context, justification, or connection to the cited reference.

Ans: We specify the article to agarwood, so the other conceptual parts are removed or modified.

Line 122: The phrase “Taxon DNA is used to calculate species identification rates” is vague and non-standard. It is unclear whether this refers to a software package or a general method.

Ans: Figure 1 added to clear the main idea of DAN barcoding and Taxon DNA.

Lines 157–185: This section lacks novelty and primarily repeats earlier content. It misses the opportunity to discuss modern techniques in product authentication and their role in regulatory oversight. The narrative should be expanded to include practical tools used in commercial or forensic settings

Ans: Repeated text/ paragraph has been removed.

Lines 200-245: The section lacks scientific rigor. While methods are mentioned (e.g., fungal or chemical induction), the authors provide no comparative data, mechanisms, or performance metrics. The discussion remains at a descriptive level without evaluating the effectiveness or limitations of each approach. Terms like "chemical inducers" or "biological agents" are used broadly, but no specific compounds or strains are named or characterized.

Ans: Biological agents like fungal strains and chemical compounds are explained in lines 224-238.

Lines 250–259: The authors appear to equate the use of fungal inoculation with "biotechnology advancements," but this is conceptually incorrect. Fungal inoculation, unless coupled with molecular tools or genetic engineering, is a biological method—not a biotechnological one.

Ans: It is corrected 256 to 265

Lines 263–283: The “Present Advancements and Prospects for the Future” section is speculative and lacks depth. It makes broad claims about AI, genetic engineering, and sustainability without providing supporting references or technical detail. Moreover, it does not synthesize earlier content or offer a concrete roadmap for future research. For example, the statement about “AI-powered models” predicting optimal conditions for agarwood resin production is speculative and cited inappropriately.

Ans: This part is explained in lines 173 -180. Also, it gives findings and key factors of using neural networks and AI models.

The referenced study (Zhu et al., 2020) concerns 3D-printed multifunctional materials and is unrelated to the topic.

Ans: So far this citation not identified in article mistakenly added is removed

Lines 363–403 (“Molecular mechanisms” section): This section is misleadingly titled. It does not present or review molecular mechanisms underlying agarwood formation in any meaningful way. Instead, it summarizes general metabolomics and transcriptomic analyses without identifying specific genes, pathways, or regulatory systems involved in resin biosynthesis. The content is superficial and tangential to the article’s focus and lacks integration with the broader narrative on induction methods or sustainable use

Ans: The explanation of agarwood formation through specific genes and pathways has been addressed via charts and Figure 4.

Figure 2: The figure is not related to molecular mechanisms as the title suggests. Instead, it presents a generic sche addressed induction triggers (e.g., stress, fungi), which merely repeats earlier content. It lacks any molecular detail (e.g., gene names, biosynthetic steps) and does not support the section’s title or claims. Its inclusion in this section is redundant and scientifically uninformative.

Ans: The Figure is removed. Another molecular mechanism image is added as Figure 4.

Lines 408–444: The section on medicinal properties and toxicities is overly descriptive and lacks adequate referencing. Claimed biological activities are not supported by experimental data, citations, or mechanistic discussion. The toxicity content is similarly superficial. A critical scientific review should provide detailed references, specify experimental contexts, and distinguish between traditional use and evidence-based pharmacology

Ans: This section is merged with the pharmacological properties section 425 to 443.

Lines 417–424: The “Pharmacological Properties” section lacks any analytical content. It merely lists proposed therapeutic effects in a table without interpretation or narrative discussion. The authors should synthesize the data, evaluate the quality and limitations of the evidence, and provide critical insight. A table without context is insufficient for a scientific review.

Ans: We really appreciate this comment; we added the interpretation and narrative discussion in lines 436-443.

Lines 425–427: The hepatoprotective property is arbitrarily separated into a standalone section titled “Other Pharmacological Effects” without justification. This content should be integrated into the main pharmacological discussion or table for consistency and clarity. As it stands, the structure appears disjointed and editorially unbalanced.

Ans: We really appreciate this comment. This section is also merged with the section “pharmacological properties”

Lines 430–444: The section titled “Safety Issues and Possible Toxicities” does not fulfill the expectations of the title. It lacks toxicological data, safety assessments, or regulatory discussion. Instead, it repeats general observations from previous sections without providing new or specific information. The title should be revised to reflect the actual content, or the section should be significantly expanded and properly referenced to address toxicity and safety.

Ans: We really appreciate this comment. The title has been changed according to the section information.

Lines 445–472: The manuscript lacks a proper conclusion. The final section, titled “Summarized insights from selected key studies,” repeats technical content from earlier sections rather than synthesizing the main findings or offering critical closing thoughts. A scientifically structured review requires a clear, concise conclusion that integrates the key themes, reflects on limitations, and proposes future research directions

Ans: We really appreciate this comment. We have revised and concise the conclusion.

Comments on the Quality of English Language

The manuscript contains frequent grammatical and punctuation errors, awkward phrasing, and unclear sentence structures, all of which impair readability. Thorough language editing by a fluent English speaker or professional service is essential prior to publication.

Ans: We have proofread the whole manuscript to remove typos and errors carefully.

Round 2

Reviewer 2 Report

Comments and Suggestions for Authors

In the revised manuscript, the authors describe barcoding in detail, but my remark was aimed at clarifying the need for barcoding for agarwood. Barcoding is one of the methods for identifying a plant species, or even a subspecies. O.K. What is the significance of the barcoding result? According to Table 3, different species of Aquilaria produce agarwood of different chemical compositions. But the authors admit that there is no standard grading system for resin quality (line 70). The manuscript does not explain the purpose for which DNA barcoding, molecular markers, and metabolomics are needed. Does agarwood from different species of Aquilaria differ in cost, importance, or use?
In addition, the authors indicate the matK, rbcL, and ITS genes for DNA barcoding (lines 106-108), but here they do not provide any references with examples of successful use of this method. In general, the description of DNA barcoding for Aquilaria species does not allow us to evaluate the success of this method for authentication of all Aquilaria species. Only some examples of identification of one specific species are given. What markers should be used for 100% identification of a plant sample to the Aquilaria species?
Table 3 (pages 12-21), in my opinion, requires a change in format. Especially considering the requirement to format references as [number]. Why write the name of the compound 4 times if authors can enter a separate column for the names of the compounds and mark their presence with “+” with a reference in the [number] format?
Section «Pharmacological properties»: Specify what applications of agarwood find in modern medicine. Are these some registered drugs? In which countries? Or is it traditional medicine? Considering the high cost of agarwood, is there any prospect for practical use of its pharmacological properties, indicated in table (again) 3 (page 24)?

The other parts of the manuscript are written well enough and do not require serious changes.

Author Response

Reviewer 2:

Comment 1: In the revised manuscript, the author describes barcoding in detail, but my remark was aimed at clarifying the need for barcoding for agarwood. Barcoding is one of the methods for identifying a plant species, or even a subspecies. O.K. What is the significance of the barcoding result? According to Table 3, different species of Aquilaria produce agarwood of different chemical compositions. But the authors admit that there is no standard grading system for resin quality (line 70). The manuscript does not explain the purpose for which DNA barcoding, molecular markers, and metabolomics are needed. Does agarwood from different species of Aquilaria differ in cost, importance, or use?

ANS 1: We appreciate the reviewer’s insightful comment and the opportunity to clarify the role of DNA barcoding, molecular markers, and metabolomics in the context of agarwood research and quality authentication.

While it is true, as stated in line 70, that there is no universally accepted grading system for agarwood resin quality due to its variation in fragrance, density, and resin content, From line 82 to 91 some parts are modified.

Comment 2: In addition, the authors indicate the matK, rbcL, and ITS genes for DNA barcoding (lines 106-108), but here they do not provide any references with examples of successful use of this method. In general, the description of DNA barcoding for Aquilaria species does not allow us to evaluate the success of this method for authentication of all Aquilaria species. Only some examples of identification of one specific species are given. What markers should be used for 100% identification of a plant sample to the Aquilaria species?

ANS 2: We thank the reviewer for pointing out the need to (i) cite concrete, peer‑reviewed examples of DNA‑barcoding success in Aquilaria and (ii) clarify which marker (or marker set) delivers reliable species‑level authentication.
Below we supply the missing citations and explain why a multi‑locus (or “super‑barcode”) strategy is currently the only way to approach “100 %” identification accuracy across the entire genus. Complete answer with citation from line 133 to 139.

Comment 3: Table 3 (pages 12-21), in my opinion, requires a change in format. Especially considering the requirement to format references as [number]. Why write the name of the compound 4 times if authors can enter a separate column for the names of the compounds and mark their presence with “+” with a reference in the [number] format?

ANS 3: Thank you for the suggestion. We believe the current format enhances readability and allows readers to clearly associate each compound with its corresponding species and reference.
Comment 1: Section «Pharmacological properties»: Specify what applications of agarwood find in modern medicine. Are these some registered drugs? In which countries? Or is it traditional medicine? Considering the high cost of agarwood, is there any prospect for practical use of its pharmacological properties, indicated in table (again) 3 (page 24)?

ANS 4: We appreciate the reviewer’s thoughtful comment and the opportunity to elaborate on the practical applications and limitations of agarwood’s pharmacological properties.

In the revised manuscript from line 555 to 576, we have clarified that most pharmacological uses of agarwood are currently confined to traditional medicine systems, particularly Traditional Chinese Medicine (TCM), Ayurveda, and Unani, where agarwood has been used for centuries to treat ailments such as asthma, digestive disorders, pain, and anxiety.

The other parts of the manuscript are written well enough and do not require serious change.

Reviewer 3 Report

Comments and Suggestions for Authors

Figure 1 is copied entirely from the article by Zhu et al., 2022 (10.1186/s13020-022-00655-y).
Line 110. "Idea taken from (Zhu et al., 2022)." The sentence is broken and does not say anything. As far as I know, for direct copying of figures from other scientific articles, authors need to obtain permission from the journal.
Line 118. The figure caption is grammatically incorrect. The source link should be provided here. Moreover, the figure caption does not clearly explain what is being shown, which limits its informational value.
Table 1. Why do the authors write “biotechnology techniques” in the table title? Which of the examples given relate to biotechnology? The title should be changed.
Table 1 is poorly integrated into the text. Provide a link to the table in all parts of the chapter that discuss the examples given in it. Otherwise, the table seems to float disconnected from the surrounding content and lacks proper contextual integration.
Lines 256, 490, and 494. The authors again persistently refer to some "biotechnological strategy advancements" in agarwood induction without providing any evidence to support this. For example, on Line 256 they refer to the work of Yan et al., 2019a (10.3390/molecules24101990). But the cited work does not mention biotechnology; the word does not appear there at all.
Figure 2 is copied entirely from the article by Ngadiran et al., 2023 (10.1016/j.biteb.2023.101337). There is no reference to the source. As far as I know, to directly copy figures from other scientific articles, authors must obtain permission from the source journal. Reproducing a figure without appropriate attribution and permission is a serious issue that may violate copyright.
Figure 4 is a simplified copy of the figure from the article by Tan et al., 2019 (10.3389/fpls.2019.00122). The source is referenced, but not in the context of this figure. This form of implicit reuse without explicit acknowledgment in the figure legend is misleading. It must be clarified that the figure is adapted from Tan et al., and permission must be obtained if it exceeds fair use. Moreover, the figure caption does not clearly explain what is being shown, which limits its informational value.
Lines 372–416 (“Molecular mechanisms” section): This section still does not present or review molecular mechanisms underlying agarwood formation in any meaningful way. The authors simply write general words, giving the example of Rhodiola imbricata for some reason.
Yes, the new figure is more relevant to the topic of the chapter, but the authors do not describe it in any way or explain its inclusion in the context. The link to the figure is given after the phrase "metabolic pathways in plants are made by the combined study of transcriptome data and metabolic profiles, as shown in Figure 4," while the figure shows the wound-induced signal transduction mechanisms for the sesquiterpene biosynthesis - as this is mentioned in the article by Tan et al., 2019 from which this diagram was taken.
The authors do not provide any specific genes, pathways, or regulatory systems involved in resin biosynthesis. The content is superficial and tangential to the article’s focus and lacks integration with the broader narrative on induction methods or sustainable use.
This section needs substantial revision. A proper review of molecular mechanisms should summarize actual transcriptomic or functional genomics studies on Aquilaria spp., rather than providing vague generalities or unrelated examples.

Comments on the Quality of English Language

The manuscript contains numerous grammatical errors, awkward phrasing, and non-academic sentence constructions that impede clarity and readability. Several sentences are incomplete or lack logical flow. Word choices are sometimes imprecise or colloquial for scientific writing, and transitions between ideas are often abrupt, weakening the coherence of the text. Additionally, the manuscript exhibits inconsistencies in tense usage, article placement, and subject-verb agreement. 

Author Response

Reviewer 3:

Comment 1:Figure 1 is copied entirely from the article by Zhu et al., 2022 (10.1186/s13020-022-00655-y).
Line 110. "Idea taken from (Zhu et al., 2022)." The sentence is broken and does not say anything. As far as I know, for direct copying of figures from other scientific articles, authors need to obtain permission from the journal.

ANS: “The design of the figure was inspired by Zhu et al., 2022, but the illustration was entirely created by the authors using original elements and modified colors.” We also modified the figure.           

Comment 1:Line 118. The figure caption is grammatically incorrect. The source link should be provided here. Moreover, the figure caption does not clearly explain what is being shown, which limits its informational value.

ANS: Figure 1. Taxonomic resolution of common plant DNA barcodes from family to subspecies level. Figure inspired by Zhu et al., 2022, and created by the authors.

Caption is corrected

Comment 2:Table 1. Why do the authors write “biotechnology techniques” in the table title? Which of the examples given relate to biotechnology? The title should be changed.
Table 1 is poorly integrated into the text. Provide a link to the table in all parts of the chapter that discuss the examples given in it. Otherwise, the table seems to float disconnected from the surrounding content and lacks proper contextual integration.

ANS: We removed the term biotechnology for typos error we changed it with suitable term hopefully it will clear the idea. ”Table 1. Overview of scientific and AI-based techniques used in the authentication and quality assessment of agarwood resin.”

Comment 3:Lines 256, 490, and 494. The authors again persistently refer to some "biotechnological strategy advancements" in agarwood induction without providing any evidence to support this. For example, on Line 256 they refer to the work of Yan et al., 2019a (10.3390/molecules24101990). But the cited work does not mention biotechnology; the word does not appear there at all.

ANS: Thank you  for your feedback we changed the caption accordingly “Organic farming, chemical induction, biological inducers, and controlled inoculation are modern methods for producing agarwood (Yan et al., 2019a).”
Comment 4:Figure 2 is copied entirely from the article by Ngadiran et al., 2023 (10.1016/j.biteb.2023.101337). There is no reference to the source. As far as I know, to directly copy figures from other scientific articles, authors must obtain permission from the source journal. Reproducing a figure without appropriate attribution and permission is a serious issue that may violate copyright.

ANS: Schematic overview of artificial and conventional methods used for agarwood resin induction. The workflow was adapted from Ngadiran et al., 2023 (https://doi.org/10.1016/j.biteb.2023.101337), but the diagram and photographs were created by the authors using original content and lab-generated images. We also modified the figure entirely.

Comment 5:Figure 4 is a simplified copy of the figure from the article by Tan et al., 2019 (10.3389/fpls.2019.00122). The source is referenced, but not in the context of this figure. This form of implicit reuse without explicit acknowledgment in the figure legend is misleading. It must be clarified that the figure is adapted from Tan et al., and permission must be obtained if it exceeds fair use. Moreover, the figure caption does not clearly explain what is being shown, which limits its informational value.

ANS: Thank you for your valuable feedback. We acknowledge that the original figure by Tan et al. (2019) inspired the conceptual framework presented in our Figure 4. However, we would like to clarify that the figure in our manuscript is not a direct copy but a newly created, simplified, and modified illustration. We have significantly altered the design by changing the layout, direction of information flow, color scheme, and graphical elements to better suit the context and focus of our study. We modified the picture entirely.

To ensure transparency and proper attribution, we have now revised the figure legend to include the statement.

Comment 6:Lines 372–416 (“Molecular mechanisms” section): This section still does not present or review molecular mechanisms underlying agarwood formation in any meaningful way. The authors simply write general words, giving the example of Rhodiola imbricata for some reason.
Yes, the new figure is more relevant to the topic of the chapter, but the authors do not describe it in any way or explain its inclusion in the context. The link to the figure is given after the phrase "metabolic pathways in plants are made by the combined study of transcriptome data and metabolic profiles, as shown in Figure 4," while the figure shows the wound-induced signal transduction mechanisms for the sesquiterpene biosynthesis - as this is mentioned in the article by Tan et al., 2019 from which this diagram was taken.
The authors do not provide any specific genes, pathways, or regulatory systems involved in resin biosynthesis. The content is superficial and tangential to the article’s focus and lacks integration with the broader narrative on induction methods or sustainable use.

This section needs substantial revision. A proper review of molecular mechanisms should summarize actual transcriptomic or functional genomics studies on Aquilaria spp., rather than providing vague generalities or unrelated examples.

ANS: To improve the scientific rigor and relevance of this section, we have revised it to now include specific genes, enzymes, and regulatory pathways known to be involved in agarwood resin formation from line 516 to 530,

Round 3

Reviewer 2 Report

Comments and Suggestions for Authors

Due to the limited time for preparing the review, I did not have time to carefully read the entire text of the revised manuscript, so this review is based on the authors' responses to my comments. In relation to my previous review, almost all major remarks have been retained with a change in format so that it is easier for the authors to respond step-by-step. And I ask the authors to respond more substantively than in the responses to my previous reviews, with the corresponding changes to the manuscript.

Major remarks:
Remark 1: In their response, the authors point out the changes in the text in lines 82-91. I did not find any explanations in this fragment of the revised manuscript, so I repeat my questions:

1) What information can the result of DNA barcoding an agarwood sample provide to a researcher with the current development of technology? Is it possible to determine the authentication of the origin, species, and quality of agarwood based on the results of barcoding?

2) Does agarwood from different Aquilaria species or from differnt origins differ in value or uses?

Remark 2: The authors have changed the text in lines 133-139, but the essence of the text remains the same, so I repeat my question for this my remark:

3) The authors list the matK, rbcL, and ITS genes for DNA barcoding but do not provide any references with examples of successful use of this method for Aquilaria species. What studies confirm that this set of markers can be successfully used to identify different Aquilaria species?

Remark 3:

4) Please format Table 3 according to IJMS requirements.

Remark 4: The authors have made changes to the description of the use of agarwood in traditional medicine. However, I have doubts about the prospects for using agarwood in modern medicine. I have the following questions regarding this remark:

5) Do the authors think that the cost and volume of production of agarwood are significant obstacles to its use as a medicine in modern medicine?

6) Which species of Aquilaria as a source of agarwood have potential for practical use of the pharmacological properties listed in Table 4?

Author Response

Comments and Suggestions for Authors

Due to the limited time for preparing the review, I did not have time to carefully read the entire text of the revised manuscript, so this review is based on the authors' responses to my comments. In relation to my previous review, almost all major remarks have been retained with a change in format so that it is easier for the authors to respond step-by-step. And I ask the authors to respond more substantively than in the responses to my previous reviews, with the corresponding changes to the manuscript.

Major remarks:
Remark 1: In their response, the authors point out the changes in the text in lines 82-91. I did not find any explanations in this fragment of the revised manuscript, so I will repeat my questions:

1) What information can the result of DNA barcoding an agarwood sample provide to a researcher with the current development of technology? Is it possible to determine the authentication of the origin, species, and quality of agarwood based on the results of barcoding?

2) Does agarwood from different Aquilaria species or from differnt origins differ in value or uses?

Response:
Thank you very much for these comments.

Agarwood from different Aquilaria species and different geographic origins can vary significantly in value, uses, and market demand. These differences are mainly due to variations in resin composition, aroma profile, and cultural preferences, which directly affect grading and pricing.

According to these suggestions, we revised the section and incorporated new information in the updated manuscript (lines 102–120 in the newly downloaded version from the journal).

Remark 2: The authors have changed the text to lines 133-139, but the essence of the text remains the same, so I repeat my question for this my remark:

3) The authors list the matK, rbcL, and ITS genes for DNA barcoding but do not provide any references with examples of successful use of this method for Aquilaria species. What studies confirm that this set of markers can be successfully used to identify different Aquilaria species?

Response:

Thank you for this comment. The current version of the manuscript does demonstrate successful DNA barcoding of Aquilaria using matK, rbcL, and ITS (see Lines 102–120 in the updated file; the earlier line numbers referenced in the review correspond to an outdated version).

Remark 3:

4) Please format Table 3 according to IJMS requirements.

Response:

Thank you very much for this comment. The table 3 has been revised and reformatted in accordance with the International Journal of Molecular Sciences (IJMS) guidelines. The updated version is now consistent with the journal’s formatting requirements for tables.

Remark 4: The authors have made changes to the description of the use of agarwood in traditional medicine. However, I have doubts about the prospects for using agarwood in modern medicine. I have the following questions regarding this remark:

5) Do the authors think that the cost and volume of production of agarwood are significant obstacles to its use as a medicine in modern medicine?

Response:

Thank you for this comment. Yes, both the high cost and the limited production volume of agarwood are significant constraints to its widespread adoption in modern medicine. Agarwood is formed through a slow and unpredictable natural process—resin accumulation in Aquilaria trees following wounding or infection—which can take several years to decades. Even with artificial induction techniques, yields remain low and variable.

From an economic standpoint, agarwood is one of the most expensive raw medicinal materials globally, with high-grade resin fetching thousands of US dollars per kilogram. This cost is driven by its rarity, labor-intensive harvesting, and limited cultivation capacity.

In modern pharmaceutical development, large-scale, standardized production of active compounds is essential for clinical trials, regulatory approval, and eventual commercialization. The scarcity and high price of agarwood resin create barriers to consistent supply, dosage standardization, and cost-effective production

Reviewer 3 Report

Comments and Suggestions for Authors
  1. Table 1 is poorly integrated into the chapter, with only a single reference to it appearing at the very end. This is insufficient, as the examples presented in the table are discussed throughout the text. Each relevant mention should include a direct reference to Table 1, allowing readers to easily consult it and better understand the context.
  2. Similarly, Figure 4 is inadequately integrated into the discussion. It is currently referenced in the sentence: "Combined analysis of transcriptomic and metabolic data may reveal functional genes as well as primary and secondary metabolic pathways in plants (Figure 4)" (Lines 358–360), yet the figure does not illustrate this concept. Furthermore, the sentence is immediately followed by: "By combining transcriptomics with metabolomics, researchers can investigate the association between genes and metabolites and clarify the molecular regulatory processes underlying metabolite dynamics (De Luca et al., 1989)" (Lines 360–363), which repeats the same idea. One of these two sentences should be removed to avoid redundancy. Additionally, citing De Luca et al., 1989 in this context is inappropriate, as integrative metabolomic-transcriptomic approaches did not exist at that time. A more recent and relevant reference within the past five years should be used instead.
  3. Moreover, Figure 4 lacks any meaningful description in the text. I recommend adding several explanatory sentences around line 372, immediately after the phrase "stressors such as fungal infection and mechanical damage," to describe the schematic in Figure 4. This would be an appropriate place to mention the stress hormones and signal transduction mechanisms involved in sesquiterpene biosynthesis, as presented in the article by Tan et al., 2019, from which the figure was adapted. A clear reference to Figure 4 should then be inserted just before the description of the mevalonate (MVA) and methylerythritol phosphate (MEP) pathways to guide the reader effectively.

Author Response

  1. Table 1 is poorly integrated into the chapter, with only a single reference to it appearing at the very end. This is insufficient, as the examples presented in the table are discussed throughout the text. Each relevant mention should include a direct reference to Table 1, allowing readers to easily consult it and better understand the context.

Response:
Thank you very much for these comments. We thank the reviewer for this helpful observation. We have revised the text so that each method or technique described in the chapter and listed in Table 1 is now directly referenced in the corresponding part of the main text also added some sentences to overview the table from line 167 to 170.

  1. Similarly, Figure 4 is inadequately integrated into the discussion. It is currently referenced in the sentence: "Combined analysis of transcriptomic and metabolic data may reveal functional genes as well as primary and secondary metabolic pathways in plants (Figure 4)" (Lines 358–360), yet the figure does not illustrate this concept. Furthermore, the sentence is immediately followed by: "By combining transcriptomics with metabolomics, researchers can investigate the association between genes and metabolites and clarify the molecular regulatory processes underlying metabolite dynamics (De Luca et al., 1989)" (Lines 360–363), which repeats the same idea. One of these two sentences should be removed to avoid redundancy. Additionally, citing De Luca et al., 1989 in this context is inappropriate, as integrative metabolomic-transcriptomic approaches did not exist at that time. A more recent and relevant reference within the past five years should be used instead.

Response:
Thank you very much for these comments. We have revised the section. In the newly downloaded version of the article from the journal, the relevant text appears at line 364. To address the reviewer’s concern, we have removed the repetitive sentence to eliminate redundancy as lines 369-386. Citing De Luca et al., 1989 is removed and new reference is added. The figure position is changed to line 370.

  1. Moreover, Figure 4 lacks any meaningful description in the text. I recommend adding several explanatory sentences around line 372, immediately after the phrase "stressors such as fungal infection and mechanical damage," to describe the schematic in Figure 4. This would be an appropriate place to mention the stress hormones and signal transduction mechanisms involved in sesquiterpene biosynthesis, as presented in the article by Tan et al., 2019, from which the figure was adapted. A clear reference to Figure 4 should then be inserted just before the description of the mevalonate (MVA) and methylerythritol phosphate (MEP) pathways to guide the reader effectively.

Response:
We appreciate this suggestion. Additionally, we have repositioned Figure 4 to line 370, placing it immediately after the discussion of fungal infection and mechanical damage, as it more appropriately illustrates this context. The difference lies in the line numbers — you mentioned line 363. Accordingly, we have added several lines from 370to 386 to illustrate the diagram and explain the functionality of the MVA and MEP pathways.

Round 4

Reviewer 2 Report

Comments and Suggestions for Authors

The authors responded to all my comments. Unfortunately, not all the authors' explanations were included in the corrected version of the manuscript. I think that adding explanations to the text would have improved the manuscript.

Column 1 of Table 3 should include the names of the compounds found in agarwood resin.

In addition, the authors should bring the manuscript formatting into line with the journal's requirements. For example, the first use of "jasmonic acid" (line 234), but the abbreviation JA is introduced only on line 375 and, for some reason, again on line 386. However, JA is not included in the list of abbreviations.

Author Response

The authors responded to all my comments. Unfortunately, not all the authors' explanations were included in the corrected version of the manuscript. I think that adding explanations to the text would have improved the manuscript.

Responses :We thank the reviewer for this observation. We have carefully revised the manuscript and incorporated the necessary explanations at relevant points in the text to improve clarity and strengthen the discussion. According to the previous suggestions (Do the authors think that the cost and volume of production of agarwood are significant obstacles to its use as a medicine in modern medicine? We added revision from lines 440-430 and (Does agarwood from different Aquilaria species or from different origins differ in value or uses?) Added revision from lines 164 to 167. All the remaining responses are included in the new version of manuscript.

Column 1 of Table 3 should include the names of the compounds found in agarwood resin.

Responses :Thank you for this suggestion. We have revised Table 3, and Column 1 now lists the names of the compounds identified in agarwood resin.

In addition, the authors should bring the manuscript formatting into line with the journal's requirements. For example, the first use of "jasmonic acid" (line 234), but the abbreviation JA is introduced only on line 375 and, for some reason, again on line 386. However, JA is not included in the list of abbreviations.

Responses :Thank you for pointing this out. We have carefully revised the manuscript to ensure consistency with the journal’s formatting requirements. The abbreviation ‘JA’ for jasmonic acid is now introduced at its first mention and used consistently thereafter. We have also included ‘JA’ in the list of abbreviations. Also, the other abbreviations are included in the abbreviation list.
